# Early Oral Administration of Ginseng Stem-Leaf Saponins Enhances the Peyer’s Patch-Dependent Maternal IgA Antibody Response to a PEDV Inactivated Vaccine in Mice, with Gut Microbiota Involvement

**DOI:** 10.3390/vaccines11040830

**Published:** 2023-04-12

**Authors:** Fei Su, Junxing Li, Yin Xue, Bin Yu, Shiyi Ye, Lihua Xu, Yuan Fu, Xiufang Yuan

**Affiliations:** 1Institute of Animal Husbandry and Veterinary Science, Zhejiang Academy of Agricultural Sciences, Hangzhou 310002, China; sufei@zaas.ac.cn (F.S.); lijunx@zaas.ac.cn (J.L.); yub@zaas.ac.cn (B.Y.); yesy@zaas.ac.cn (S.Y.); xulihua@zaas.ac.cn (L.X.); fuy@zaas.ac.cn (Y.F.); 2Zhejiang Center of Animal Disease Control, Hangzhou 310020, China; schroyen@163.com

**Keywords:** GSLS, PEDV, mucosal immune enhancement, maternal IgA, Peyer’s patches, gut microbiota

## Abstract

Neonatal piglets during the first week of life are highly susceptible to porcine epidemic diarrhoea virus (PEDV) infection, with mortality rates reaching 80–100%. Passive lactogenic immunity remains the most effective way to protect neonates from infection. Although safe, inactivated vaccines provide little or no passive protection. Here, we administered ginseng stem-leaf saponins (GSLS) to mice before parenteral immunization with an inactivated PEDV vaccine to investigate the effect of GSLS on the gut–mammary gland (MG)–secretory IgA axis. Early oral GSLS administration potently increased PEDV-specific IgA plasma cell generation in the intestine, facilitated intestinal IgA plasma cell migration to the MG by enhancing the chemokine receptor (CCR)10-chemokine ligand (CCL)28 interaction, and ultimately promoted specific IgA secretion into milk, which was dependent on Peyer’s patches (PPs). Additionally, GSLS improved the gut microbiota composition, especially increasing probiotic abundance, and these microflora members promoted the GSLS-enhanced gut–MG–secretory IgA axis response and were regulated by PPs. In summary, our findings highlight the potential of GSLS as an oral adjuvant for PEDV-inactivated vaccines and provide an attractive vaccination strategy for lactogenic immunity induction in sows. Further studies are required to evaluate the mucosal immune enhancement efficacy of GSLS in pigs.

## 1. Introduction

Porcine epidemic diarrhoea virus (PEDV), which belongs to the family Coronaviridae, causes watery diarrhoea, dehydration and reduced reproductive performance in adult pigs and up to 100% mortality in neonatal piglets, leading to millions of dollars lost annually among swine producers [1]. The induction of localized immune responses, especially the production of intestinal secretory immunoglobulin A (IgA) antibodies, is critical for preventing PEDV attachment and invasion [2]. Due to the immature immune system of neonatal piglets, passive lactogenic immunity remains the most common and effective strategy to protect neonates from PEDV infection [3]. Maternal antibodies, primarily the IgA isotype, provide local passive protection to the piglet intestinal tract [3,4]. IgA class-switched B cells are mainly generated in gut-associated lymphoid tissue (GALT), including Peyer’s patches (PPs) and mesenteric lymph nodes (MLNs), which acquire gut-homing properties and migrate to the lamina propria of the intestine, where they mature into IgA antibody-secreting plasma cells [5]. During lactation, IgA plasma cells stimulated in the gut following virulent PEDV infection traffic to the mammary gland (MG) and produce PEDV-specific secretory IgA antibodies in the milk, which is defined as the gut–MG–secretory IgA axis [6,7]. Enhancing the trafficking of PEDV-specific IgA-producing cells from the intestine to the MG and the accumulation of specific IgA antibodies in milk are strategies for vaccination to protect suckling piglets from PEDV immediately after birth. As previously described in mouse models, the migration of IgA plasma cells from the gut to the MG is mediated mostly by the surface chemokine receptor (CCR) 10 interacting with chemokine ligand (CCL) 28 during lactation [3,8,9]. The blockade of CCL28 or deletion of CCR10 severely suppresses the accumulation of IgA antibodies in the MG, resulting in a lack of ingested IgA in suckling neonates [10,11]. Although these studies have not been replicated in swine, there is evidence that CCL28 is also highly expressed in the MG of swine during lactation, and the increase in CCL28 in the MG is accompanied by the accumulation of IgA plasma cells over the course of gestation [3]. These similarities support using mice as an immune model. However, due to possible species differences in lymphoid structure between mice and swine, further research is needed to corroborate the results of mouse studies. In sow herds, oral inoculation with live virulent PEDV or infectious material such as faeces and gut tissues collected from PEDV-infected piglets in the same herd could provide a high rate of protection with an increased titre of specific IgA antibodies in colostrum and milk [3,12]. However, the vertical transmission of PEDV via colostrum was detected in infected sows on pig farms, which resulted in piglet infection [13]. Considering the risk of live vaccines and controlled oral exposure, inactivated vaccines have a better safety profile but are degraded by the gastrointestinal environment when delivered via the oral route. Moreover, inactivated vaccine immunization through parenteral injection mainly elicits the production of PEDV-specific IgG antibodies that are dominant in serum and colostrum, and their levels in milk decline rapidly, providing little lactogenic immunity to piglets [3]. Therefore, the development of safe and optimized vaccine regimens for gestating sows is crucial to boost the gut–MG–secretory IgA axis and protect neonates from infection. One alternative solution is to develop an effective adjuvant for a PEDV-inactivated vaccine that targets the gut–MG axis.

Total ginseng saponins isolated from the stem-leaf of *Panax ginseng* C. A. Meyer (GSLS) have a pharmacological activity similar to those from the root, and they can be produced at a much lower price [14]. Many studies have reported the mucosal adjuvant property of GSLS for enhancing intestinal immunity in response to various vaccines in animals. For example, Zhai et al. found that when compared with the vaccine alone, the additional administration of GSLS via the oral route improved the effect of an infectious bursal chicken disease vaccine on the production of intestinal intraepithelial lymphocytes (iIELs) and IgA plasma cells in the lamina propria of the chicken duodenum, jejunum and ileum, which provide effective protection [15]. Similarly, Li et al. found that the early oral administration of GSLS promoted the production of iIELs and IgA plasma cells in the duodenum of mice induced by a foot-and-mouth disease vaccine [16]. More importantly, our previous study demonstrated that the oral administration of nanoparticle-encapsulated GSLS prior to intramuscular inoculation of PEDV could activate CD11b^−^CD103^+^ dendritic cells in the draining MLNs and increase the IgA plasma cells in the intestine, thereby enhancing specific IgA and neutralization antibody responses in the local intestine of mice [14]. These immune enhancement activities are related to the chemical structure of saponins [17]. The carbohydrate portion of saponins may promote the phagocytosis of antigen-presenting cells (APCs) by interacting with surface receptors, while the acyl chain domains are responsible for delivering antigens to APCs and enhancing adaptive immune responses. However, the effect of GSLS on the gut–MG axis remains unclear. In addition, resident microbes in the intestine, including *Lactobacillus*, initiate immune activities in the GALT to induce the class switching of B lymphocytes to IgA plasmablasts that further mature to IgA plasma cells and secrete IgA antibodies [18,19]. Oral administration of ginseng saponin monomers was found to regulate the homeostasis of the gut microbiota and host–gut metabolism in animals [20,21,22]. However, the underlying mechanisms of how ginseng saponins regulate the interaction between the mucosal immune system and intestinal flora remain unclear.

In the present study, we administered GSLS before parenteral immunization with an inactivated PEDV vaccine to preactivate the intestinal mucosal immune system and investigate the effect of GSLS on the gut–MG axis. Simultaneously, we utilized 16S rRNA sequencing technology to study the regulatory effect of GSLS on the gut microbiota. On this basis, we further used broad-spectrum antibiotics (ABX) to disturb the intestinal microbiota and investigated the role of the intestinal flora on the enhanced maternal IgA antibody response by GSLS, which was further confirmed by a faecal microbiota transplantation (FMT) experiment. Moreover, we established a mouse model of PP deficiency to explore the role of PPs in the process by which GSLS enhance the maternal IgA response and improve the intestinal microbial composition, which was further verified in severe combined immunodeficiency (SCID) mice that received PP-derived cells. Our findings reveal for the first time that the early oral administration of GSLS can significantly enhance the maternal IgA response to a PEDV-inactivated vaccine, and this effect is dependent on PPs and is related to the PP-regulated intestinal microbiota.

## 2. Materials and Methods

### 2.1. Cells and Virus

Vero cells were kindly donated by Prof. Guangzhi Tong from the Shanghai Veterinary Research Institute of the Chinese Academy of Agricultural Sciences and cultured in Dulbecco’s modified Eagle’s medium (DMEM, Life Technologies, Shanghai, China) supplemented with 8% (*v*/*v*) foetal bovine serum (FBS, Thermo Fisher Scientific, Shanghai, China), 100 U/mL of penicillin and 100 µg/mL of streptomycin (Genomcell, Hangzhou, China) in an incubator with 5% CO_2_ at 37 °C. The PEDV ZJ-ZX2018-C10 strain (GenBank No. MK250953) was propagated and titrated in Vero cells at 1 × 10^6.8^ TCID_50_ (50% tissue culture infectious dose)/mL. The virus was then inactivated using 0.05% β-propiolactone (Solarbio, Beijing, China) at 4 °C for 12 h and another 2 h at 37 °C and used as a PEDV vaccine. Inactivation was confirmed by blindly passaging the treated virus three times in Vero cells.

### 2.2. Adjuvant

GSLS were obtained from Hongjiu Biotechnology Co., Ltd. (Jilin, China). Solutions were sterilized through a 0.22 µm filter, and the endotoxin level was less than 0.5 EU/mL.

### 2.3. Animal Studies

BALB/c, ICR (C.B-17/ICR-+/+Jcl) and ICR mice with severe combined immunodeficiency (C.B-17/ICR-scid/scidJcl, SCID) were obtained from the Shanghai Laboratory Animal Centre (SLAC, Shanghai, China) and maintained under controlled conditions. For the initial evaluation of adjuvanticity, 6- to 8-week-old female BALB/c mice (*n* = 5/group) were gavaged daily with GSLS (5 mg/kg/day) or saline for 2 weeks. After the final administration, the mice were intramuscularly given 1 × 10^5.8^ TCID_50_ of inactivated PEDV vaccine twice at an interval of 3 weeks. Mice given saline alone were set as negative control. One week before the booster immunization, the mice were mated with healthy males and euthanized on day 7 after parturition. Blood and small intestine (SI) samples were harvested for antibody quantification. Serum were obtained by centrifugation at 1000× *g* for 10 min. Three-centimetre sections of the SI were extracted and homogenized in 400 µL of PBS containing 5% FBS and 0.02% sodium azide. The supernatant was harvested after centrifugation. Mammary glands were collected for flow cytometry, histochemistry, immunofluorescence and transcriptomic analysis. Faecal pellets were collected the day before vaccination and on day 6 after parturition for 16S rRNA sequencing. Milk samples were collected from the stomach contents of suckling mice on day 7 after birth. The stomachs were homogenized in PBS (10 μL per mg), and the supernatant was collected by centrifugation. To disturb the intestinal microbiota, 3- to 4-week-old female BALB/c mice were given a mixture of antibiotics (1 g/L ampicillin, 1 g/L neomycin, 0.5 g/L metronidazole, and 0.5 g/L vancomycin) in their drinking water for 4 weeks; this was followed by another 2 weeks of daily oral treatment with GSLS or saline before PEDV vaccination. In some experiments, FMT was performed once daily from days 4 to 10 after antibiotic treatment. A total of 3–4 fresh faecal pellets harvested from healthy mice who were continuously gavaged with GSLS or saline for 2 weeks were homogenized in PBS (500 µL) and were given to each female mouse (aged 6–8 weeks) via oral gavage (200 µL per mouse). Female BALB/c mice lacking PPs (PP-null mice) were generated by intravenous injection of 600 µg of anti-mouse IL-7R antibody (GTX01473, GeneTex, Irvine, CA, USA) on embryonic day 14 [23,24] and were orally administered GSLS or saline once daily for 2 weeks before vaccination. In addition, six- to eight-week-old female wild-type (C.B-17/ICR-+/+Jcl) and C.B-17/ICR-scid/scidJcl mice were gavaged daily with GSLS or saline for 2 weeks. C.B-17/ICR-scid/scidJcl mice received mononuclear cells isolated from the PPs of wild-type mice via intravenous injection 1 day after the first gavage (1 × 10^6^ cells per mouse). All mice (*n* = 5/group) were vaccinated and mated in the same way as described above and were sacrificed on day 7 after parturition. Various organs and tissues were collected and analysed.

### 2.4. Enzyme-Linked Immunosorbent Assay (ELISA) for Antibody Quantification

ELISA was used to determine anti-PEDV IgG and IgA antibody responses. Briefly, 96-well microplates precoated with the spike protein of PEDV were from anti-PEDV antibody ELISA kits (HomSun, Hangzhou, China). The plates were added with 100 µL of diluted samples (serum diluted in 1:100, intestine diluted in 1:2 and milk dilutedin 1:2) and incubated at 37 °C for 1 h. After a round of washing, the plates were added with 100 µL of goat anti-mouse IgG (1:5000, BK-M050, BIOKER, Hangzhou, China) or IgA (1:2000, sc-3791, Santa Cruz Biotechnology, Shanghai, China) conjugated with horseradish peroxidase (HRP) and incubated at 37 °C for 1 h. After another round of washing, 100 µL of tetramethylbenzidine substrate solution (IDEXX, Westbrook, ME, USA) was added to each well, and the reaction was stopped using a stop solution after 15 min. The optical density (OD) of the wells was read at a wavelength of 450 nm with Multiskan MK3 (Thermo Fisher Scientific, Shanghai, China).

### 2.5. Histochemistry and Immunofluorescence

MG and SI tissues were fixed in 4% paraformaldehyde (Solarbio, Beijing, China) and embedded in paraffin. For immunohistochemical analysis, endogenous peroxidase activity was inhibited by incubation with 3% H_2_O_2_ for 30 min, and nonspecific antigens were blocked with rabbit serum for 30 min. The sections (4 µm) were then incubated sequentially with a goat anti-mouse IgA antibody (1:1000, ab97231, Abcam, Waltham, MA, USA) and HRP-conjugated rabbit anti-goat IgG (1:100, A21030, Abbkine, Wuhan, China) overnight at 4 °C. The signal was developed with 3,3′-diaminobenzidine (DAB) (BOSTER, Wuhan, China). Finally, the sections were counterstained with haematoxylin. For immunofluorescence analysis, the sections were incubated with specific primary antibodies overnight at 4 °C. The following antibodies were used: a goat anti-mouse IgA antibody (1:1000, ab97231, Abcam, Waltham, MA, USA), anti-CCR10 polyclonal antibody (1:100, PA1-21617, Invitrogen, Waltham, MA, USA) and anti-CCL28 polyclonal antibody (1:100, DF7045, Affinity Biosciences, Shanghai, China). The sections were then incubated with fluorescence-labelled secondary antibodies (Invitrogen, Waltham, MA, USA). Nuclei were stained with DAPI (Roche, Shanghai, China). Images were documented using a Pannoramic SCAN (3D Histech). Representative images were acquired using CaseViewer software. Five fields of each slide were randomly selected. The histochemistry score (H-score) of each field and the fluorescence intensity of IgA staining were analysed. Haematoxylin-eosin staining was also performed to investigate the development of iIELs during the experimental period. The number of cells in three fields on each slide was calculated to perform statistical analysis.

### 2.6. Cell Isolation and Flow Cytometry

MG tissues and PPs were digested with 0.2% collagenase IV (Sangon Biotech, Shanghai, China) at 37 °C for 1 h to obtain mononuclear cells. Red blood cells (RBCs) were lysed with RBC lysis buffer (Solarbio, Beijing, China). For surface staining, cells were incubated and stained with the Fixable Viability Dye eFluor 506/780 Sample Pack Kit (eBioscience, Waltham, MA, USA) and fluorochrome-conjugated antibodies. For intracellular staining, a Fixation/Permeabilization Kit (MultiSciences, Hangzhou, China) was used. The anti-mouse antibodies used in this study included PerCP-Cy5.5-labeled anti-CD3 (145-2C11), FITC-labelled anti-CD45R (B220) (RA3-6B2) and PE-labelled anti-IgA (mA-6E1). The PerCP-Cy5.5-labeled anti-CD3 and FITC-labelled anti-CD45R antibodies were purchased from MultiSciences (Hangzhou, China), and the PE-labelled anti-IgA antibody was obtained from eBioscience (Waltham, MA, USA). The analysis was performed by a FACS Canto™ (BD Biosciences, San Diego, CA, USA) and FlowJo (v10.0).

### 2.7. Transcriptome Analysis

All technical manipulations were performed by Novogene Bioinformatics Technology Co., Ltd. (Beijing, China). Total RNA was extracted using an RNeasy Mini Kit (Qiagen, MD, USA) according to the manufacturer’s instructions. RNA quality and quantity were checked by the RNA Nano 6000 Assay Kit of the Bioanalyzer 2100 system (Agilent Technologies, CA, USA) and NanoPhotometer^®^ spectrophotometer (IMPLEN, Westlake Village, CA, USA). Sequencing libraries were constructed using the NEBNext^®^ UltraTM RNA Library Prep Kit (New England Biolabs, Ipswich, MA, USA) following the manufacturer’s instructions. RNA-seq experiments were performed on an Illumina HiSeq platform using a paired-end read length of 2 × 150 bp.

### 2.8. Quantitative Real-Time PCR (qPCR) Analysis

Total RNA was extracted from the MG and SI tissues using Animal Tissue Total RNA Extraction Kits (Simgen, Hangzhou, China), and synthesis of cDNA was performed by reverse transcription with PrimeScript™ RT Master Mix (Takara, Shiga, Japan). qPCR analysis was performed using TB Green^®^ Premix Ex Taq™ II (Tli RNaseH Plus) (Takara). The following primers were used: β-actin forward 5′-CAAGGACCTCTACGCCAACAC-3′ and reverse 5′-TGGAGGCGCGATGATCTT-3′; CCR10 forward 5′-AGAGCTCTGTTACAAGGCTGATGTC-3′ and reverse 5′-CAGGTGGTACTTCCTAGATTCCAGC-3′; APRIL forward 5′-GGTGGTATCTCGGGAAGGAC-3′ and reverse 5′-CCCCTTGATGTAAATGAAAGACA-3′; TACI forward 5′-CCAGGATTGAGGCTAAGTAGCG-3′ and reverse 5′-GGGGAGTTTGCTTGTGACC-3′; and CCL28 forward 5′-GATTGTGACTTGGCTGCTGTC-3′ and reverse 5′-TATGGTTGTGCGGGCTGA-3′. Gene expression levels were normalized to that of β-actin and are presented as fold induction relative to the control according to the 2^−ΔΔCT^ method.

### 2.9. Gut Microbial Analysis

All technical manipulations were performed by Novogene Bioinformatics Technology Co., Ltd. (Beijing, China). The genomic DNA of faeces was extracted using the Magnetic Soil and Stool DNA Kit (TIANGEN Biotech, Beijing, China) according to the manufacturer’s guidelines. The integrity and size of DNA were checked on a 1% agarose gel, and DNA concentrations and purity were determined by a NanoPhotometer^®^ spectrophotometer. The hypervariable V3-V4 region of the bacterial 16S ribosomal RNA (rRNA) was amplified with the primers 341F (5′-CCTAYGGGRBGCASCAG-3′) and 806R (5′-GGACTACNNGGGTATCTAAT-3′) by a T100TM PCR thermal cycler (Bio-Rad, Hercules, CA, USA). The PCR product was extracted from a 2% agarose gel and purified using the GeneJET Gel Extraction Kit (Thermo Fisher Scientific). After quantification, the purified amplicons were pooled in equimolar concentrations and paired-end sequenced on an Illumina NovaSeq platform according to the standard protocols by Novogene Bioinformatics Technology (Beijing, China).

### 2.10. Statistical Analysis

For quantitative analysis, the results were processed by an unpaired two-tailed Student’s t test (for two groups) and ordinary one-way ANOVA with Tukey’s test (for three or more groups) using GraphPad Prism 7.0. Data are expressed as the mean ± standard deviation (SD). A *p* value < 0.05 was considered to indicate a statistically significant difference. For transcriptome analysis, the paired-end clean reads were aligned to the reference genome using Hisat2 software (v2.0.5). The mapped reads of each sample were assembled by StringTie (v1.3.3b) with a reference-based approach. The feature Counts (v1.5.0-p3) were used to count the number of reads mapped to each gene. The DESeq2 R package (1.20.0) was employed to analyse the differentially expressed genes (DEGs). The resulting *p* values were adjusted using Benjamini and Hochberg’s method for controlling the false discovery rate. The genes that exhibited both an adjusted *p* value < 0.05 and |log2(fold change)| > 1 in the comparisons were considered DEGs. For gut microbial analysis, the paired-end reads were demultiplexed, quality-filtered and processed using the Quantitative Insights into Microbial Ecology (QIIME) open-source software package [25]. Sequences with more than 97% similarity were assigned to the same operational taxonomic unit (OTU), and chimeric sequences were identified and removed. The taxonomy of each OTU representative sequence was analysed using Ribosomal Database Project (RDP) Classifier against the 16S rRNA database (Silva v138) with a confidence threshold of 0.7. At the OTU level of genotypes, beta diversity was calculated using weighted UniFrac distances and visualized through principal coordinate analysis (PCoA) plots.

## 3. Results

### 3.1. GSLS Enhance the PEDV-Specific Maternal IgA Antibody Response

First, we sought to evaluate the immune enhancement effect of GSLS on a PEDV-inactivated vaccine (Figure 1A). Antibody quantification results showed that the mice orally gavaged with GSLS before vaccination exhibited higher titres of PEDV-specific IgA in SI and milk when compared with the mice immunized with the vaccine only, but PEDV-specific IgG antibody titres were equivalent in serum, indicating that the intestinal mucosa and MG are the main action sites of GSLS after oral administration (Figure 1B). Parenteral vaccination alone increased only serum IgG levels, not intestinal or milk IgA levels. The intestinal mucosal immune system is considered the first line of defense against PEDV, and lactogenic immunity at an early age plays a crucial role in the passive protection of neonatal suckling piglets from infection [26]. The maternal antibody response was further confirmed by immunohistochemistry, immunofluorescence and flow cytometry analysis. As shown in Figure 1C–G, after oral administration of GSLS, the plasma cells secreting IgA accumulated significantly in the MG at a level around three times higher than those in the vaccine-only group. In addition, transcriptome analysis of the MG showed that GSLS treatment promoted the production of IgA in the intestinal immune network compared to that with vaccine-only immunization, which implicated the enhancement effect of GSLS on the gut–MG–secretory IgA axis (Figure 1H). The data on related genes are presented in Appendix A. The qPCR analysis confirmed that the expression of CCR10 (Ccr10), a proliferation-inducing ligand (APRIL, Tnfsf13), and its receptor transmembrane activator and calcium-modulator and cyclophilin interactor (TACI, Tnfrsf13b) in the MG was significantly upregulated by GSLS (Figure 1I). The gut-homing receptor CCR10 is required for the migration of IgA plasma cells to the MG, and APRIL and TACI are critical for the differentiation of B cells into plasma cells [27]. Furthermore, as shown in the transcriptome results, GSLS treatment significantly promoted the expression of tight junction-related genes in the MG compared to that in the vaccine-only group, which suggested the regulatory effect of GSLS on the integrity of the mammary epithelial barrier and lactation function.

### 3.2. GSLS Improve the Composition of the Gut Microbiota

Considering that the intestinal microbiota may play a significant role in the maintenance of intestinal immune homeostasis, we next explored the regulatory effect of GSLS on the intestinal microflora. As shown in Figure 2A, beta-diversity analysis of gut microbiome signatures showed significant differences between the mice treated with GSLS and those treated with saline, and this discrepancy was not affected by parenteral vaccination. Among the microorganisms detected, the relative abundance of *Lactobacillus*, *Ligilactobacillus* and *Odoribacter* in the intestinal flora of the mice treated with GSLS was dramatically higher than that in the mice given saline (Figure 2B,C). Notably, the microbial structure improvement by GSLS was maintained for approximately 6 weeks from pre-vaccination to post-partum, indicating that oral administration of GSLS has a long-term regulatory effect on the intestinal microbiota.

### 3.3. Intestinal Microflora Constituents Participate in the GSLS-Enhanced Specific Maternal IgA Response

We next investigated the role of the intestinal microflora on the GSLS-enhanced specific intestinal and mammary IgA responses (Figure 3A). As shown in Figure 3B,C, the usage of ABX significantly altered the microbial diversity. Compared with that in the mice without ABX, the abundance of *Lactobacillus*, *Ligilactobacillus* and *Odoribacter* regulated by GSLS was obviously lower in the mice given ABX, suggesting that ABX suppress the regulation of the intestinal flora by GSLS (Figure 3D). Moreover, in the vaccine-only group, the abundance of *Lactobacillus* but not *Ligilactobacillus* or *Odoribacter* was decreased as a result of ABX. More importantly, due to the disturbance of the intestinal microflora by ABX, the enhanced generation of IgA plasma cells in the PPs and the production of PEDV-specific IgA antibody and CCR10 in SI by GSLS was significantly reduced (Figure 3E–H). This result was consistent with that observed in the MG, where the population of IgA plasma cells and the expression of CCR10/CL28 were dramatically decreased, leading to a diminished secretion of specific IgA antibodies in milk (Figure 3I–N). Notably, although the enhancement effect of GSLS on the gut–MG–secretory IgA axis was significantly impaired by ABX, it was still stronger than that in the vaccine-only group.

We further performed faecal microbiota transplantation (FMT) using faeces from healthy mice given GSLS or saline. Specifically, the recipient mice were pretreated with ABX for 4 weeks, followed by FMT and vaccination (Figure 4A). We found that the microbial diversity apparently changed after FMT (Figure 4B,C). The abundances of *Lactobacillus* and *Ligilactobacillus* were obviously higher in the mice given faeces from GSLS-treated donors than in the mice who received GSLS or faeces from saline-treated donors (Figure 4D). However, FMT was incapable of restoring the abundance of *Odoribacter*, which was sensitive to ABX. After receiving faeces from GSLS-treated donors, the mice showed an increased number of IgA plasma cells in their PPs, as well as an increased PEDV-specific IgA antibody titre and CCR10 expression in the SI compared to those in the mice given GSLS or faeces from saline-treated donors (Figure 4E–H). Similar results were also found in the MG, where the population of IgA plasma cells and the expression of CCR10/CL28 were significantly promoted, resulting in an increased secretion of specific IgA antibodies in the milk (Figure 4I–N). In addition, even though the abundance of *Ligilactobacillus* was increased in the gut after FMT, parenteral vaccination alone could not promote either intestinal or mammary immunity.

### 3.4. PPs Play a Key Role in GSLS Regulation of the Gut Microbiota and Enhancement of the Maternal IgA Response

PPs, the major organized lymphoid tissues of the SI, are the dominant source of IgA-producing cells [28,29]. To investigate whether PPs direct the regulatory effect of GSLS on the microbial milieu and maternal IgA response, we generated PP-deficient (PP-null) mice and performed experiments (Figure 5A). Due to the lack of PPs, the microbial diversity was changed (Figure 5B,C). Under the same GSLS treatment regimen, the abundance of *Lactobacillus*, *Ligilactobacillus* and *Odoribacter* was obviously lower in the PP-null mice than in the normal mice, indicating that PPs are indispensable for the GSLS regulation of the intestinal microflora (Figure 5D). However, the deletion of PPs reduced the abundance of only *Ligilactobacillus* but not *Lactobacillus* or *Odoribacter* in the vaccine-alone group. More importantly, due to the deficiency of PPs, GSLS failed to promote the generation of IgA plasma cells and IgA antibodies in the SI and MG, which was accompanied by a decreased expression of CCR10 and CCL28 in these tissues (Figure 5E–L). Moreover, PP deficiency also decreased the population of IgA-producing cells and the expression of CCR10 and CCL28 in the SI and MG of the mice that received the vaccine alone, which exhibited a poor PEDV-specific IgA response.

We further confirmed these findings by transferring cells isolated from the PPs of wild-type mice into SCID mice, which lack both mature T and B cells (Figure 6A). As shown in Figure 6B–D, compared with that of the wild-type mice, the microbial diversity of SCID mice was significantly reduced, and the abundance of *Lactobacillus*, *Ligilactobacillus* and *Odoribacter* was seriously decreased, which was not improved by GSLS treatment. After cell transplantation, GSLS significantly increased the abundance of *Lactobacillus* and *Ligilactobacillus* but not *Odoribacter*, which indicated that the regulation of intestinal flora by GSLS requires PP-derived cells. However, this approach did not restore the abundance of *Odoribacter* via transplantation of the PP cells. The GSLS-treated SCID mice did not exhibit enhanced intestinal or mammary immune responses until they received PP cells. After cell transplantation, the population of iIELs and IgA plasma cells in the SI was increased, with a significantly stronger effect in the GSLS-treated mice than in the vaccine-only mice (Figure 6E–G). Moreover, higher levels of PEDV-specific IgA antibody and CCR10 expression were also observed in the SI of the GSLS-treated SCID mice after cell transplantation when compared with those of the mice in the vaccine-only group (Figure 6H,I). Similar results were also found in the MG of SCID mice that received cell transfer, in which the number of IgA-producing cells and the expression of CCR10/CL28 were significantly promoted by GSLS, resulting in an increased secretion of PEDV-specific IgA antibody in milk when compared to that of the vaccine-only group (Figure 7A–E). Moreover, the population of IgA plasma cells and the expression of CCR10 and CCL28 were also increased in the SI and MG of the mice that received the vaccine alone after transplantation of PP-derived cells.

## 4. Discussion

Passive lactogenic immunity to PEDV induced via the gut–MG–secretory IgA axis in sows remains a promising and effective way to protect neonates from infection. However, since PEDV can be transmitted from infected sows to neonatal piglets via colostrum, the inoculation of sows with live vaccines or the use of feedback methods is not ideal for controlling PEDV infection [13]. Although the inactivated vaccine is safe, it provides little to no passive protection. In this study, parenteral immunization with a PEDV-inactivated vaccine mainly induced the production of specific IgG antibodies in serum rather than secretory IgA in the intestine or milk, which is consistent with previous reports [3,7]. Secretory IgA plays an important role in early defence and viral containment. Using an effective mucosal adjuvant to improve the local immune status before vaccination would be a potential solution to this problem. Our previous study demonstrated that the early oral administration of GSLS could enhance the antigen-specific immune responses in the intestine, wherein GSLS could encourage the dendritic cells to take antigens to the MLNs, which further triggers the expression of gut-tropism receptors on the lymphocytes and initiates gut homing activity [14,30]. Similarly, in this study, we also found that the oral administration of GSLS before parenteral vaccination significantly increased the population of IgA-producing cells in the PPs and promoted the PEDV-specific IgA antibody response and CCR10 expression in the SI. CCR10 is vital for the migration of gut-originated IgA plasma cells to the MG [31]. As expected, the expression of CCR10 and CCL28 in the MG was increased 5–6 times upon early treatment with GSLS, followed by a more than doubled accumulation of IgA plasma cells in the MG. These findings suggest that the oral administration of GSLS could promote gut-originated PEDV-specific IgA plasma cells migrating to the MG through the interaction between the surface CCR10 and the mammary CCL28 at the initial of lactation signals, resulting in a significant increase in the secretion of PEDV-specific IgA in milk.

PPs, a major component of gut-associated lymphoid tissue (GALT), serve as the principal location for B cell differentiation into IgA plasmablasts that further mature to IgA plasma cells distributed in the lamina propria of intestine [23,32]. After GSLS treatment, the change in the population of IgA plasma cells in the MG of both normal mice and mice with intestinal bacterial disorders was positively correlated with that in PPs. One hypothesis is that PPs may be the primary source of GSLS-enriched IgA plasma cells in the lactating MG. In PP-deficient mice, GSLS failed to increase the generation of IgA-producing cells and the expression of CCR10 in the SI, which was accompanied by the suppressed expression of CCR10 and CCL28 and a sharply decreased population of IgA-producing cells in the MG, resulting in the elimination of PEDV-specific IgA antibodies in the milk. This result indicates that the GSLS enhancement of the gut–MG–secretory IgA axis relies on PPs. Further study using SCID mice that are severely deficient in functional B and T lymphocytes also confirmed this result. In SCID mice, GSLS were unable to activate the gut–MG axis. After the transplantation of PP-derived cells, which are mainly lymphocytes, GSLS stimulated the recruitment of these cells to the intestinal mucosa, enriched the CCR10-imprinted IgA plasma cells, and promoted the trafficking of IgA plasma cells to the MG to secrete PEDV-specific IgA antibody into milk, thus restoring the enhancement effect on the gut–MG–secretory IgA axis of the SCID mice. These results clearly demonstrated that PPs are the origin of enriched IgA-producing cells in the MG upon GSLS treatment and are essential for GSLS to increase the secretion of PEDV-specific IgA in milk. Furthermore, the deficiency of PPs also decreased the generation of IgA-producing cells in the SI and reduced the migration of IgA plasma cells to the MG in the mice that received vaccine alone, and these mice exhibited a poor PEDV-specific IgA response. Moreover, SCID mice that received the vaccine alone showed an increased population and enhanced migration of IgA plasma cells in the gut–MG axis after the transplantation of PP cells. These results suggested that PPs are the primary source of IgA plasma cells in the MG during lactation and are responsible for producing specific and nonspecific maternal IgA antibodies.

Gut microbiota constituents and their metabolites, such as short-chain fatty acids and bile acids, strongly promote IgA plasma cell differentiation in the intestine [33]. A lack of gut microbial stimulation results in fewer IgA plasma cells and reduced IgA production in the intestine [34,35]. The oral intake of GSLS seems to have a long-term regulatory effect on the gut microbiota, and the improved microbial structure can be maintained for more than 6 weeks. GSLS significantly increased the abundance of probiotic bacteria, such as *Lactobacillus*, *Ligilactobacillus* and *Odoribacter*, which confer health benefits to the host, such as antimicrobial activity, intestinal homeostasis maintenance and immune system stimulation [36,37,38,39]. These probiotics might further reach the intestines of newborns via colostrum and milk and subsequently improve the intestinal microenvironment of piglets, which is of great significance for reinforcing the intestinal defence of piglets against infection [39,40]. Surprisingly, parenteral immunization with the PEDV-inactivated vaccine had little impact on the structure of the intestinal microflora. Furthermore, treatment with ABX seriously impaired the GSLS enhancement of the gut–MG–secretory IgA axis, which was observed through the decreased generation of IgA plasma cells in the PPs and fewer IgA plasma cells migrating to the MG, leading to a significant loss of PEDV-specific IgA in the milk. These aberrant responses could be rescued in the recipients who received the faeces from the GSLS-administered donors, which indicated that the intestinal microflora is involved in the process of GSLS enhancement of the gut–MG-secretory IgA axis. PP deficiency caused a dramatic change in the gut microbial diversity, especially inhibiting the regulation of the probiotic strains by GSLS. The SCID mice exhibited abnormalities in the gut microbial composition, with extremely low diversity, which could not be improved by GSLS. In the SCID mice, it was not until after transplantation of PP-derived cells that GSLS restored the intestinal flora, indicating that the regulatory effect of GSLS on the intestinal flora depends on PPs. Of note, the GSLS enrichment of *Odoribacter* was susceptible to ABX and PP deficiency, and *Odoribacter* abundance could not be easily rescued by FMT and PP cell transplantation. Moreover, in the mice with a disturbed intestinal microbiota, GSLS still enhanced the specific maternal IgA antibody response when compared with those after vaccine-only treatment, although it was weaker than that in normal mice, indicating that in addition to the intestinal flora, there are other important factors, such as PPs affecting the ability of GSLS to enhance the specific maternal IgA antibody response. Based on these and previous results, we confirm that PPs are indispensable for the GSLS enhancement of the PEDV-specific maternal IgA antibody response, and we hypothesize that GSLS may enhance the gut–MG–secretory IgA axis through two pathways: 1) by directly regulating the immune cells in the PPs to generate IgA plasma cells and promote the trafficking of these cells to the MG to secrete PEDV-specific IgA into milk and 2) by regulating the interaction between the intestinal flora and the immunocytes in the PPs to indirectly boost the specific IgA response in the gut–MG axis.

## 5. Conclusions

In conclusion, we found that the early oral administration of GSLS effectively enhanced the gut–MG–secretory IgA axis in response to a PEDV-inactivated vaccine by activating intestinal immunity, boosting the generation of PEDV-specific IgA plasma cells in the SI, and facilitating the migration of enterogenic IgA plasma cells to the MG through CCR10/CCL28 interaction, thereby promoting the secretion of specific IgA into milk, which is dependent on PPs. In addition, GSLS improved the composition of the gut microbiota, especially by increasing the abundance of probiotics, and these bacteria were involved in the process whereby GSLS strengthens the gut–MG–secretory IgA axis and were regulated by PPs. Our results provide a better understanding of the adjuvant effect of GSLS on PEDV-inactivated vaccines, and they stimulate the development of vaccination strategies for the induction of lactogenic immunity in sows.

## Figures and Tables

**Figure 1 vaccines-11-00830-f001:**
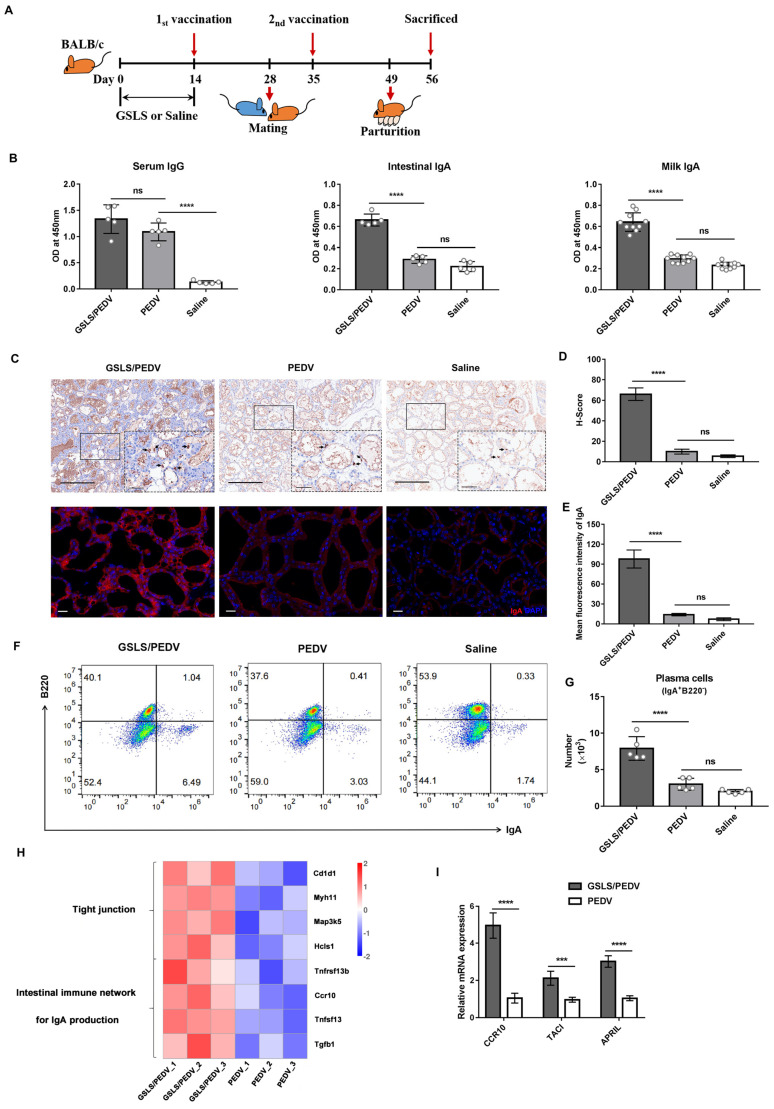
GSLS enhance the maternal IgA antibody response to a PEDV inactivated vaccine. Female BALB/c mice (*n* = 5/group) were gavaged daily with GSLS (5 mg/kg/day) or saline for 2 weeks, and were subsequently intramuscularly given 1 × 10^5.8^ TCID_50_ of an inactivated PEDV vaccine twice at an interval of 3 weeks. One week before the booster immunization, the mice were mated with healthy males and euthanized on day 7 post-partum. (**A**) Experiment scheme. (**B**) The levels of PEDV-specific serum IgG and intestinal and milk IgA were measured by ELISA. Milk samples were harvested from the stomach of suckling mice (*n* = 10/group) on day 7 after birth. (**C**) Representative immunohistochemistry (IHC) and immunofluorescence (IF) images of IgA plasma cells in the mammary gland. Scale bar = 250, 50 µm for IHC and 20 µm for IF. (**D**,**E**) Quantitative analysis of IgA accumulation. (**F**,**G**) IgA plasma cells in the mammary gland was measured by flow cytometry. (**H**) Transcriptomic analysis of the mammary gland. (**I**) The mRNA expression of CCR10, TACI and APRIL in the mammary gland was determined by qPCR. Data are presented as the means ± SDs and were analysed by one-way ANOVA with Tukey’s multiple comparisons (**B**,**D**,**E**,**G**) and Student’s *t* test (**I**). Abbreviations: GSLS, ginseng stem−leaf saponins; PEDV, porcine epidemic diarrhoea virus. *** *p* < 0.001, **** *p* < 0.0001. ns, no significant difference.

**Figure 2 vaccines-11-00830-f002:**
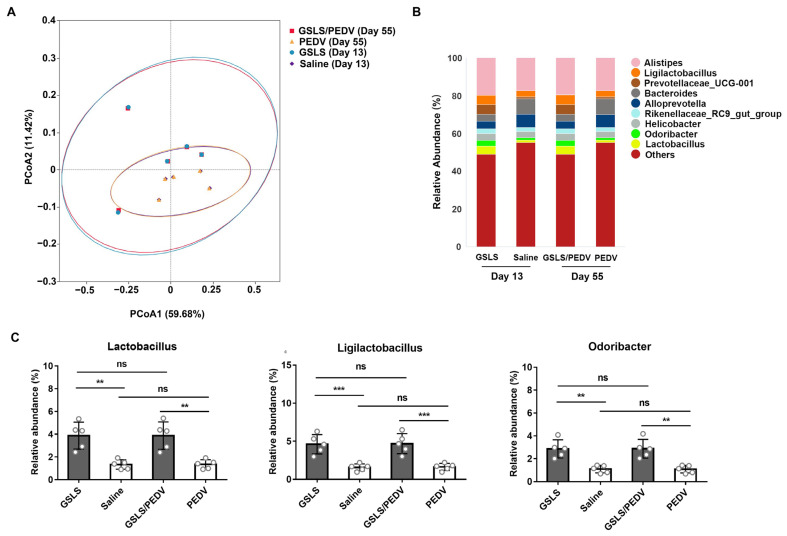
GSLS increase the abundance of beneficial bacteria in the intestinal microbiota. Female BALB/c mice (*n* = 5/group) were processed as described in Figure 1. Faecal pellets were collected the day before vaccination and on day 6 after parturition for 16S rRNA sequencing. (**A**) Principal coordinates analysis (PCoA) of the intestinal flora genera based on weighted UniFrac distances. (**B**) Relative abundance of intestinal microflora constituents at the genus level. (**C**) Representative intestinal bacteria enriched by GSLS treatment. Data are presented as the means ± SDs and were analysed by one-way ANOVA with Tukey’s multiple comparisons. Abbreviations: GSLS, ginseng stem-leaf saponins; PEDV, porcine epidemic diarrhoea virus. ** *p* < 0.01, *** *p* < 0.001. ns, no significant difference.

**Figure 3 vaccines-11-00830-f003:**
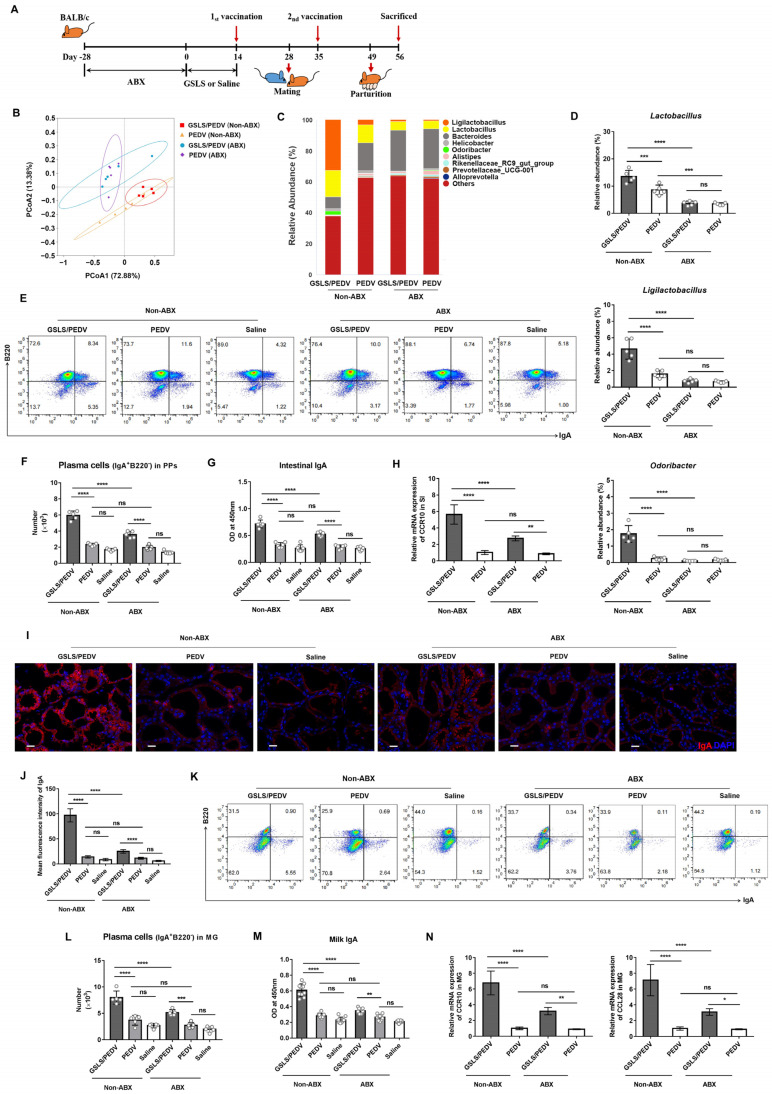
The intestinal microflora affects the GSLS-enhanced maternal IgA antibody response. After disruption of the original intestinal flora using ABX for 4 weeks, female BALB/c mice (*n* = 5/group) were processed as described in Figure 1 and euthanized on day 7 post-partum. (**A**) Experiment scheme. (**B**) PCoA of the microflora genera based on weighted UniFrac distances. (**C**) Relative abundance bar chart of intestinal flora constituents at the genus level. (**D**) Relative abundance of *Lactobacillus*, *Ligilactobacillus* and *Odoribacter* in the intestinal microflora. (**E**,**F**) The number of IgA plasma cells in the Peyer’s patches (PPs). (**G**) The level of PEDV-specific intestinal IgA. (**H**) The mRNA expression of CCR10 in the small intestine (SI). (**I**,**J**) Representative images of immunofluorescence-labelled IgA plasma cells (red) in the mammary gland (MG) and quantitative analysis. Scale bar = 20 µm. (**K**,**L**) The population of IgA plasma cells in the MG. (**M**) The OD value of PEDV-specific milk IgA. Milk samples were harvested from the stomach contents of suckling mice (n = 10/group) on day 7 after birth. (**N**) The mRNA expression of CCR10 and CCL28 in the MG. Data are presented as the means ± SDs and were analysed by one-way ANOVA with Tukey’s multiple comparisons. Abbreviations: GSLS, ginseng stem-leaf saponins; PEDV, porcine epidemic diarrhoea virus; ABX, broad-spectrum antibiotics. * *p* < 0.05, ** *p* < 0.01, *** *p* < 0.001, **** *p* < 0.0001. ns, no significant difference.

**Figure 4 vaccines-11-00830-f004:**
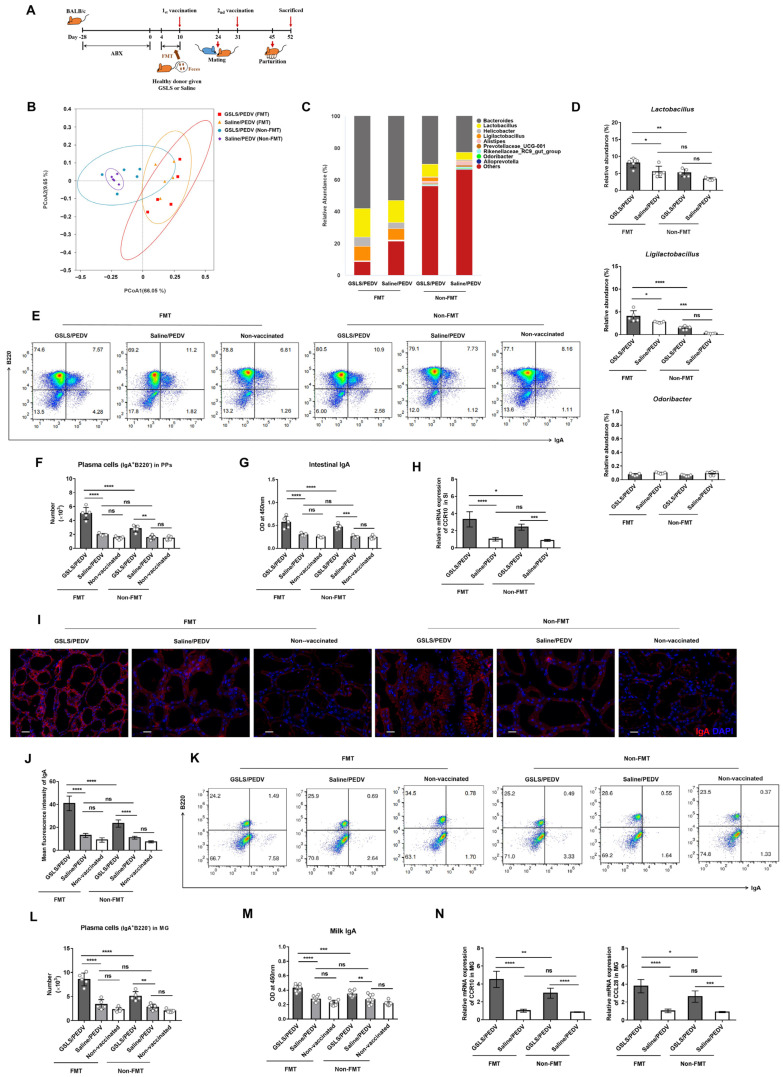
Faecal microbiota transplantation (FMT) with faeces from GSLS-treated mice promotes the maternal IgA antibody response to a PEDV-inactivated vaccine. Faeces from saline- or GSLS-treated mice were continuously given orally for 7 days to recipient female mice with a disturbed intestinal flora. The recipients (*n* = 5/group) were then vaccinated and mated as described in Figure 1 and sacrificed on day 7 post-partum. (**A**) Experiment scheme. (**B**) PCoA plots of the intestinal microflora genera based on weighted UniFrac distances. (**C**) Relative abundance of intestinal flora constituents at the genus level. (**D**) Frequencies of *Lactobacillus*, *Ligilactobacillus* and *Odoribacter* in the intestinal microflora. (**E**,**F**) The population of IgA plasma cells in PPs. (**G**) The OD value of PEDV-specific intestinal IgA. (**H**) The mRNA expression of CCR10 in the SI. (**I**,**J**) Representative images of IgA plasma cells (red) in the MG and quantitative analysis. Scale bar = 20 µm. (**K**,**L**) The population of IgA plasma cells in the MG. (**M**) The OD value of PEDV-specific milk IgA. Milk samples were collected from the stomach of suckling mice (n = 10/group) on day 7 after birth. (**N**) The mRNA expression of CCR10 and CCL28 in the MG. Data are presented as the means ± SDs and were analysed by one-way ANOVA with Tukey’s multiple comparisons. Abbreviations: GSLS, ginseng stem-leaf saponins; PEDV, porcine epidemic diarrhoea virus; FMT, faecal microbiota transplantation; PPs, Peyer’s patches. * *p* < 0.05, ** *p* < 0.01, *** *p* < 0.001, **** *p* < 0.0001. ns, no significant difference.

**Figure 5 vaccines-11-00830-f005:**
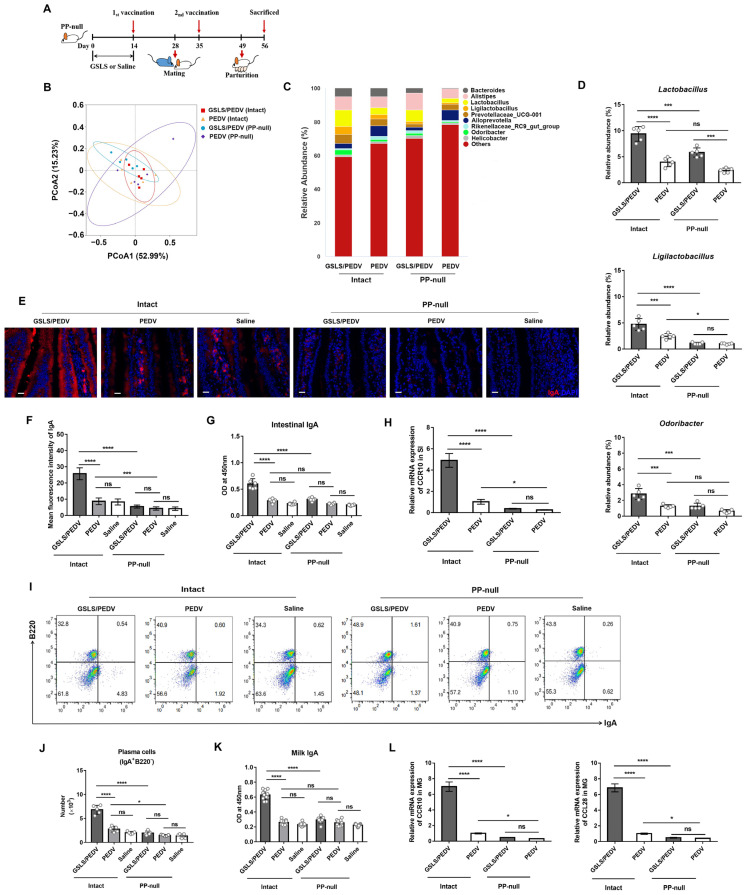
The improvement of the intestinal microbiota and maternal IgA antibody response by GSLS depends on PPs. Normal and PP-null mice (*n* = 5/group) were processed as described in Figure 1 and euthanized on day 7 post-partum. (**A**) Experiment scheme. (**B**) PCoA plots of the intestinal microflora genera based on weighted UniFrac distances. (**C**) Relative abundance chart of intestinal flora constituents at the genus level. (**D**) Frequencies of *Lactobacillus*, *Ligilactobacillus* and *Odoribacter* in the intestinal microflora. (**E**,**F**) Representative images of IgA plasma cells (red) in the SI and quantitative analysis. Scale bar = 20 µm. (**G**) The OD value of PEDV-specific intestinal IgA. (**H**) The mRNA expression of CCR10 in the SI. (**I**,**J**) The population of IgA plasma cells in the MG. (**K**) The OD value of PEDV-specific milk IgA. Milk samples were collected from the stomach of suckling mice (*n* = 10/group) on day 7 after birth. (**L**) The mRNA expression of CCR10 and CCL28 in the MG. Data are presented as the means ± SDs and were analysed by one-way ANOVA with Tukey’s multiple comparisons. * *p* < 0.05, *** *p* < 0.001, **** *p* < 0.0001. ns, no significant difference.

**Figure 6 vaccines-11-00830-f006:**
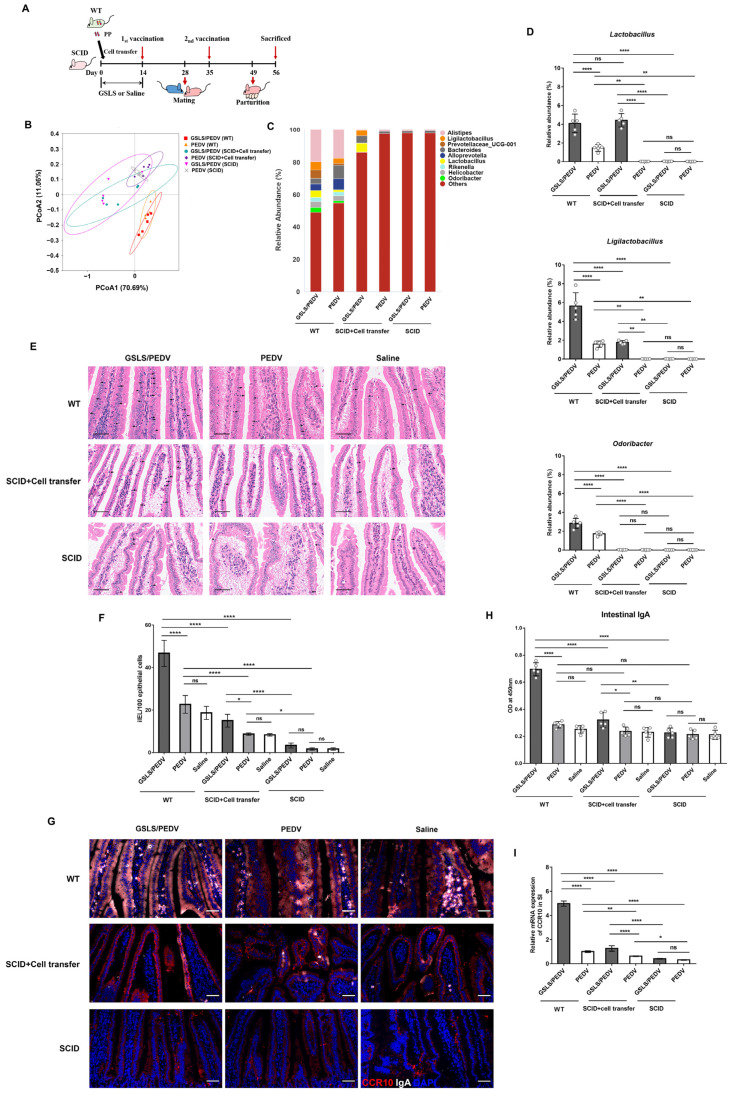
Immunodeficient mice transplanted with PP cells re-establish the intestinal microbiota and regain intestinal immunity after GSLS treatment. Female wild-type (C.B-17/ICR-+/+Jcl, WT) and immunodeficient (C.B-17/ICR-scid/scidJcl, SCID) mice were gavaged daily with GSLS or saline for 2 weeks. Some SCID mice received PP cell transplantation 1 day after the first gavage. All mice (*n* = 5/group) were vaccinated and mated in the same way as described in Figure 1 and were sacrificed on day 7 after parturition. (**A**) Experiment scheme. (**B**) PCoA of the intestinal microflora genera based on weighted UniFrac distances. (**C**) Relative abundance bar chart of intestinal flora constituents at the genus level. (**D**) Frequencies of *Lactobacillus*, *Ligilactobacillus* and *Odoribacter* in the intestinal microflora. (**E**,**F**) Representative haematoxylin-eosin staining images of intestinal intraepithelial lymphocytes (iIELs) (indicated as arrows) in the SI and quantitative analysis. Scale bar = 50 µm. (**G**) Representative images of immunofluorescence-labelled IgA (white) and CCR10 (red) in the SI. Scale bar = 50 µm. (**H**) The production of PEDV-specific intestinal IgA. (**I**) The mRNA expression of CCR10 in the SI. Data are presented as the means ± SDs and were analysed by one-way ANOVA with Tukey’s multiple comparisons. Abbreviations: WT, wide-type; SCID, severe combined immunodeficiency; OD, optical density; PP, Peyer’s patch. * *p* < 0.05, ** *p* < 0.01, **** *p* < 0.0001. ns, no significant difference.

**Figure 7 vaccines-11-00830-f007:**
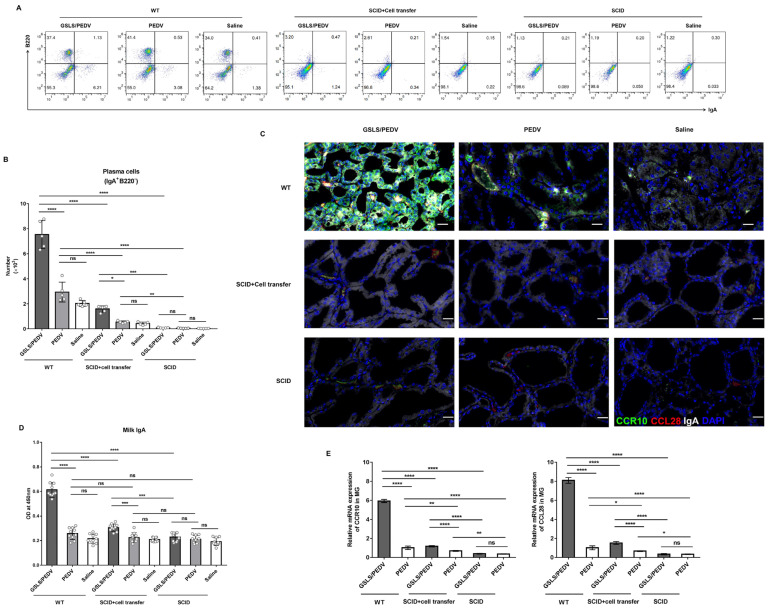
Immunodeficient mice with PP cells retain the maternal IgA antibody response after GSLS administration. Mice (*n* = 5/group) were grouped and processed as described in Figure 6. (**A**) Representative flow cytometry plot of IgA-producing cells from the MG. (**B**) The population of IgA plasma cells in the MG. (**C**) Representative images of immunofluorescence-labelled IgA (white), CCR10 (green) and CCL28 (red) in the MG. Scale bar = 20 µm. (**D**) The level of PEDV-specific milk IgA. Milk samples were harvested from the stomach of suckling mice (*n* = 10/group) on day 7 after birth. (**E**) The mRNA expression of CCR10 and CCL28 in the MG. Data are presented as the means ± SDs and were analysed by one-way ANOVA with Tukey’s multiple comparisons. * *p* < 0.05, ** *p* < 0.01, *** *p* < 0.001, **** *p* < 0.0001. ns, no significant difference.

## Data Availability

All data generated or analysed during the current study are available from the corresponding author on reasonable request.

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
