# Peer review of "Early Oral Administration of Ginseng Stem-Leaf Saponins Enhances the Peyer’s Patch-Dependent Maternal IgA Antibody Response to a PEDV Inactivated Vaccine in Mice, with Gut Microbiota Involvement"

_vaccines, 2023, doi:10.3390/vaccines11040830_

Round 1

Reviewer 1 Report

In this study Su et al. used ginseng stem-leaf saponins (GSLS) to enhance the maternal IgA response to a PEDV inactivated vaccine by activating intestinal immunity, facilitating the migration of intestinal IgA plasma cells to the mammary gland and promoting the secretion of PEDV-specific IgA antibody into milk. The authors confirmed that this immune enhancement of GSLS depends on Peyer's patches and is related to intestinal flora. The paper is well written, and the work is important not only for PEDV research, but also for other vaccine development fields.

Minor concerns:

- This study involved several experiment schemes, please indicate clearly in the figure panels, including immunization and sampling plans.

 -Please specify the efficiencies of the qPCR assays.

 -Line 323, 324: The letters in bracket are lowercase, which are inconsistent with the previous ones. Please unify them into uppercase.

 -To show enhanced PEDV specific antibody response by GSLS, authors should have determined the titer of antibody in different groups by serial dilution of sample instead of using ELISA OD values from single dilution.

-Line 89: Strain name should be italicized.

Author Response

Minor concerns:

  1. This study involved several experiment schemes, please indicate clearly in the figure panels, including immunization and sampling plans.

A: Schematic diagrams for the design of animal experiments have been added in figures, and relevant figure legends have been modified.

  1. Please specify the efficiencies of the qPCR assays.

A: The efficiencies of β-actin, APRIL, CCL28, CCR10 and TACI are 106.75%, 111.88%, 120.93%, 107.47% and 101.80%, respectively.

  1. Line 323, 324: The letters in bracket are lowercase, which are inconsistent with the previous ones. Please unify them into uppercase.

A: Have been modified (Line 331-332).

  1. To show enhanced PEDV specific antibody response by GSLS, authors should have determined the titer of antibody in different groups by serial dilution of sample instead of using ELISA OD values from single dilution.

A: We optimized the sample dilution in the preliminary test.

  1. Line 89: Strain name should be italicized.

A: Strain name has been italicized (Line 97).

Reviewer 2 Report

Your research was adequately performed.

Author Response

Thank you very much for taking time to review this manuscript!

Reviewer 3 Report

The study by Su et al. is based on a mouse model and evaluates the ability of GSLS as a potential immune adjuvant to enhance the immunogenicity and immune responses of PEDV inactivated vaccine from the aspects of lgA production, antibody levels, and chemokine expression. Furthermore, this study reveals the mechanism of GSLS in enhancing maternal lgA immunity, the results showed GSLS plays a role in the intestinal-mammary gland-lgA axis, stimulate the production of PEDV-specific lgA plasma cells by changing the proportion of intestinal flora, and increase the expression of chemokine receptors in the mammary gland to promote the transport of lgA plasma cells from the intestine to the mammary gland. Peyer's patches play an indispensable key role in the whole process.

The Lactogenic immunity of sows is vital for the control of PEDV infection. This study discussed the safety and effectiveness of PEDV vaccines, and introduced adjuvants is important in induce robust immune repsonses. The study focused on the interaction between the host immune system and intestinal flora, revealing the possible mechanism of immune adjuvant action. The experiments are well-designed, and the hypothesis was demonstrated using different animal models to control variables, reflecting the rigor of scientific research. The results are important for PEDV vaccine development. There are several concerns, which if addressed, could strengthen the manuscript.

Major comments:

1.       The title and abstract should clearly indicate that this study is based on the mouse model, and the introduction should compare the similarities and differences between mice and pigs in the intestinal-mammary gland-lgA axis to support the rational of using mice as model to study PEDV immunity.

2.       In the results and discussion section, quantitative analysis should be conducted on parameters such as the number of positive cells and mRNA levels, such as how many folds they are upregulated and whether it is significant.

3.       In Fig2. What is the exact time point the fecal samples for intestinal microbiota analysis was collected? Samples from both time points are needed to demonstrate the durability of the effect.

4.       The group names in Fig. 4 are not clear. If “GSLS/PEDV” represent the group mice were subjected to FMT before vaccination using fecal pellets from mice treated with GSLS, then another group need to be named “Saline/PEDV”, while the “Saline” group in image DEFHIJKL that represent mice were not vaccinated should be named “non-Vaccinated”.

5.       In Materials and Method 2.4: Please explain what antigen was used in ELISA, protein/whole virus?

6.       Please provide a brief procedure of PPs transplantation in “Materials and Methods” section.

7.       There is repetition in the introduction and discussion sections, such as in lines 79-83 and lines 493-497, please refine them.

Minor comments

8.       For better understanding by the readers, it is suggested to create a schematic diagram for the design of animal experiments, which includes the timepoints for pre-treatment, immunization, and sample collection.

9.       Since Odoribacter showed no significant difference in the FMT experiment, considering whether it is because gastric acid may inactivate the probiotics during gavage?

10.   Please discuss the possible mechanisms of upregulating CCR10 and CCR28 expression in mammary glands after oral administrate GSLS.

Author Response

Major comments:

  1. The title and abstract should clearly indicate that this study is based on the mouse model, and the introduction should compare the similarities and differences between mice and pigs in the intestinal-mammary gland-lgA axis to support the rational of using mice as model to study PEDV immunity.

A: “Mouse model” has been indicated in the title (Line 4) and abstract (Line 16). The rational of using mice as model to study PEDV immunity has been illustrated in the introduction (Line 55-61).

  1. In the results and discussion section, quantitative analysis should be conducted on parameters such as the number of positive cells and mRNA levels, such as how many folds they are upregulated and whether it is significant.

A: Have been modified in Lines 302, 310, 523-525 as recommended.

  1. In Fig2. What is the exact time point the fecal samples for intestinal microbiota analysis was collected? Samples from both time points are needed to demonstrate the durability of the effect.

A: The exact time points for sample collection have been labeled in Fig.2.

  1. The group names in Fig. 4 are not clear. If “GSLS/PEDV” represent the group mice were subjected to FMT before vaccination using fecal pellets from mice treated with GSLS, then another group need to be named “Saline/PEDV”, while the “Saline” group in image DEFHIJKL that represent mice were not vaccinated should be named “non-Vaccinated”.

A: Have been modified in Fig. 4 as recommended.

  1. In Materials and Method 2.4: Please explain what antigen was used in ELISA, protein/whole virus?

A: The coated antigen is spike protein of PEDV, and has been added in Line 175.

  1. Please provide a brief procedure of PPs transplantation in “Materials and Methods” section.

A: The procedure of PP transplantation has been described in Lines 165-170.

  1. There is repetition in the introduction and discussion sections, such as in lines 79-83 and lines 493-497, please refine them.

A: Have been modified in Line 515-518.

Minor comments:

  1. For better understanding by the readers, it is suggested to create a schematic diagram for the design of animal experiments, which includes the timepoints for pre-treatment, immunization, and sample collection.

A: Schematic diagrams for the design of animal experiments have been added in Figures.

  1. Since Odoribacter showed no significant difference in the FMT experiment, considering whether it is because gastric acid may inactivate the probiotics during gavage?

A:It’s possible. Odoribacter may be more sensitive to gastric acid or hydrolase than Lactobacillus and Ligilactobacillus.

  1. Please discuss the possible mechanisms of upregulating CCR10 and CCR28 expression in mammary glands after oral administrate GSLS.

A: The discussion has been added as recommended (Line 525-528).
